# Effects of Unilateral Neuromuscular Electrical Stimulation with Illusionary Mirror Visual Feedback on the Contralateral Muscle: A Pilot Study

**DOI:** 10.3390/ijerph20043755

**Published:** 2023-02-20

**Authors:** Xin Ye, Daniel Vala, Hayden Walker, Victor Gaza, Vinz Umali, Patrick Brodoff, Nathan Gockel, Masatoshi Nakamura

**Affiliations:** 1Department of Rehabilitation Sciences, University of Hartford, West Hartford, CT 06117, USA; 2Institute for Human Movement and Medical Sciences, Niigata University of Health and Welfare, 1398 Shimami-cho, Kita-ku, Niigata 950-3198, Japan

**Keywords:** electrical stimulation, cross-education, contralateral, neuromuscular, mirror visual feedback

## Abstract

We aim to examine the cross-education effects of unilateral muscle neuromuscular electrical stimulation (NMES) training combined with illusionary mirror visual feedback (MVF). Fifteen adults (NMES + MVF: 5; NMES: 5, Control: 5) completed this study. The experimental groups completed a 3-week NMES training on their dominant elbow flexor muscle. The NMES + MVF group had a mirror placed in the midsagittal plane between their upper arms, so a visual illusion was created in which their non-dominant arms appeared to be stimulated. Baseline and post-training measurements included both arms’ isometric strength, voluntary activation level, and resting twitch. Cross-education effects were not observed from all dependent variables. For the unilateral muscle, both experimental groups showed greater strength increases when compared to the control (isometric strength % changes: NMES + MVF vs. NMES vs. Control = 6.31 ± 4.56% vs. 4.72 ± 8.97% vs. −4.04 ± 3.85%, *p* < 0.05). Throughout the training, even with the maximally tolerated NMES, the NMES + MVF group had greater perceived exertion and discomfort than the NMES. Additionally, the NMES-evoked force increased throughout the training for both groups. Our data does not support that NMES combined with or without MVF induces cross-education. However, the stimulated muscle becomes more responsive to the NMES and can become stronger following the training.

## 1. Introduction

The cross-education effect refers to a phenomenon that training (skill or strength) on one side of the limb muscle (upper or lower limb) can improve the skill or strength of the muscle on the contralateral (untrained) side. Even though the cross-education effect has been documented for over a century [1], the underlying mechanisms of this phenomenon remain unclear. For example, a possible mechanism (a cortical mechanism) [2] is the involvement of interhemispheric connections between the two hemispheres of the brain via the corpus callosum, that training one limb can lead to changes in the activity of the opposite hemisphere [3], which may contribute to improved performance in the untrained limb. One possible application of this phenomenon can be found in the field of exercise and rehabilitation, where one may train the intact side if the opposite homologous (contralateral) limb cannot be trained due to injuries or diseases (e.g., surgery-induced immobilization, stroke). However, as a relatively novel tool, mixed findings have been reported in recent literature [4,5,6,7]. One big challenge to moving this research forward is identifying the specific exercise mode that can induce the largest magnitude of the contralateral cross-education effect.

Mirror therapy or illusionary mirror visual feedback (MVF) was first invented to treat amputees who suffered from phantom limb pain [8]. It is also a valuable strategy to help enhance motor recovery in post-stroke hemiparesis [9]. Briefly, mirror therapy creates a reflective illusion of an affected limb using a mirror placed in an individual’s midsagittal plane, facing toward the contralateral intact limb. With this setup, the affected limb is covered. The individual then looks into the mirror on the side with the unaffected contralateral limb making “mirror symmetric” movement or muscle contractions. It has been hypothesized that mirror training can augment cross-education during stringing training [10]. Recent evidence also showed that this intervention augments the cross-education of strength in normal healthy [11] and functional improvements in stroke participants [12,13].

Neuromuscular electrical stimulation (NMES) is a useful rehabilitation method for diverse clinical and special populations, as it can hinder or delay the decline of skeletal muscle strength and mass [14,15]. Using electrodes linked to a current generator, NMES administers sequences of electrical impulses to the outer layer of skeletal muscles. Unlike voluntary muscle contractions that follow the “Size Principle”, which involves the activation of small to large motor units in an orderly manner, the electrical impulses can elicit forceful involuntary muscle contractions by enlisting high-threshold motor units, thereby activating fast-twitch muscle fibers [16]. Therefore, such a unique motor unit recruitment pattern during NMES can cause greater metabolic cost and provoke greater neuromuscular fatigue compared with voluntary exercises, thereby potentially inducing different neural adaptations when compared to traditional voluntary muscle contractions. A recent scoping review [17] indicated that chronic unilateral NMES tends to significantly increase motor performance (e.g., strength, muscle excitation) of the contralateral limb. According to Hortobágyi et al. [18], electrical stimulation can activate Group II afferents, which may have excitatory effects on contralateral homologous muscles. This was confirmed in their experiment, showing stimulation training group had greater cross-education than the voluntary training group. However, a recent research study compared the cross-education effects among motor imagery, submaximal NMES, and control, but the cross-education effects were only observed in the motor imagery group [19]. The authors, therefore, suggested that cross-education does not necessarily occur if only activating the muscle (rather than activating the cortical motor regions).

Considering that both the MVF and NMES are potent stimuli on the neuromuscular system, the combined effects of these two on the exercised muscle and the non-exercised contralateral muscle is still unknown. Therefore, the purpose of this pilot study is to examine the effects of a three-week unilateral upper limb muscle (elbow flexor) neuromuscular electrical stimulation (NMES) training combined with MVF on the contralateral untrained muscle neuromuscular properties.

## 2. Materials and Methods

### 2.1. Experimental Design

To examine the potential cross-education effects of the combined unilateral NMES with MVF on contralateral muscle neuromuscular properties, a between-subject randomized controlled design (NMES combined with MVF [NMES + MVF]; NMES only [NMES]; control group [CON]) was used for this pilot study. Subjects assigned to the experimental groups completed a 3-week unilateral elbow flexor muscle NMES training (3 times per week); only the subjects in the NMES + MVF group had the MVF during the NMES training sessions. Subjects assigned to the CON group did not receive any training but remained their daily activities during the 3-week period. Baseline pre- and post-testing included maximal voluntary isometric contraction with twitch interpolation, where both arms’ isometric strength, voluntary activation level, and resting twitch were recorded and calculated. All training was done on the dominant arm based on the subjects’ throwing preferences. Thus, the potential cross-education effects were examined in the non-dominant arm. Before the initial testing session, a familiarization session was carried out to acquaint the participants with all the testing procedures. Furthermore, during this visit, the experimental group participants’ NMES training amplitude was established.

### 2.2. Subjects

A total of 15 individuals (6 women) completed this study (NMES + MVF group: 3 men and 2 women, Age = 21 ± 3 years, Height = 176.3 ± 3.4 cm, Weight = 83.9 ± 19.1 kg; NMES group: 3 men and 2 women, Age = 21 ± 2 years, Height = 168.6 ± 7.5 cm, Weight = 80.1 ± 14.5 kg; CON group: 3 men and 2 women, Age = 21 ± 3 years, Height = 174.0 ± 8.9 cm, Weight = 79.4 ± 18.0 kg). As part of the consent procedure, the study personnel distributed a questionnaire on health and exercise history to all participants to confirm the absence of any ongoing or recent (within the last six months) neuromuscular conditions or musculoskeletal upper limb injuries (including shoulders, elbows, wrists, and fingers). All participants provided their signature on a consent form prior to undergoing any experimental testing or procedures. Moreover, they received instructions to maintain their regular daily routine, including dietary intake, hydration, and sleep, while refraining from any strenuous physical activities and resistance exercises throughout the study duration. This research received approval from the University Institutional Review Board (Protocol ID#: PRO20211115) and was carried out according to the Declaration of Helsinki policy statement on the use of human subjects.

### 2.3. Procedures

#### 2.3.1. Familiarization Visit

At the onset of the visit, the participants’ standing height and body weight were measured first. After that, the rating of perceived exertion (RPE) scale, which assessed the intensity of the subjects’ muscle effort, and the visual analog scale (VAS), which measured their discomfort level, were explained to the subjects. Subsequently, they received instructions on performing the elbow flexion isometric strength test in an upright seated position using the custom-designed testing table and equipment. With the elbow joint angle at 90 degrees and the forearm upright (see details of elbow flexion isometric strength testing from Jeon et al. [20]), the subjects rehearsed contracting their elbow flexor muscles against the load cell (Model SSM-AJ-500; Interface, Scottsdale, AZ, USA), which was unmovable, for several repetitions, applying roughly 50% of their maximal effort. This was succeeded by executing two to three maximal contractions. After the practice session, the study personnel initiated the procedure for determining the twitch interpolation amplitude. Firstly, both the bicep brachii muscle bellies of the participants were wiped with alcohol pads and shaved using a razor to remove the surface hair. Once the skin was prepared, two stimulating electrodes (2 × 2-inch square TENS Unit Pads, AUVON Inc., Peachtree Corners, GA, USA) were placed above the skin over the proximal belly (cathode) and the distal tendon (anode) of the biceps brachii muscle based on the setup from Ye et al. [21]. The stimulating electrodes were connected to a high-voltage (maximal voltage 400 V) constant-current stimulator (Digitimer model DS7R; Hertfordshire, UK). With the isometric strength testing setup, the subjects were requested to relax their elbow flexor muscles (while maintaining the isometric strength testing position, i.e., 90-degree elbow joint angle with the forearm upright) during the twitch interpolation amplitude determination process using the isometric strength testing setup. The research personnel began by administering a series of stimuli (paired pulses at 100 Hz, 200 μs pulse-width) at 50 mA for each participant, increasing the intensity by 20 mA every 20 s. They continued this procedure until the twitch force reached a plateau, after which it decreased over two consecutive stimulations [22].

The last part of this visit was to determine the NMES training amplitude (only for the experimental groups). With the same setup as in Ye et al. [23], the research staff used a trigger signal generator (High Precision DDS Signal Generator Counter, Koolertron, Hong Kong, China) to control the stimulator to deliver the NMES (1 ms, 100 Hz, biphasic square waveform) to the biceps brachii muscle of the subject’s dominant arm. The researchers began with an amplitude of zero mA and gradually increased it (approximately 1 mA per second for most participants). During this time, the subjects were requested to relax their muscles as much as possible and let the investigators know when they experienced significant discomfort. Each subject completed at least three 10-s NMES trials, and the highest amplitude they achieved was documented as the maximum tolerable NMES amplitude for NMES training.

#### 2.3.2. Experimental Visits

About one week after the familiarization, the subjects returned to the lab for the baseline testing. Tests for dependent variables were conducted in the following order: non-dominant elbow flexion maximal isometric strength with twitch interpolation and dominant elbow flexion isometric strength with twitch interpolation. These procedures the calculated both arms’ isometric strength, voluntary activation level, and resting twitch force.

If assigned to the experimental groups, the subjects immediately started the first NMES training session (either NMES + MVF group or MNES group) following the baseline testing. The training lasted 3 weeks with 3 training sessions per week. During each training session, the NMES was performed on the dominant arm’s biceps muscle. Each training session consisted of twenty 20-s maximally tolerated NMES interspaced by 40 s of rest. The NMES + MVF group had a square-shaped mirror (30 cm × 46 cm) placed in their midsagittal plane, facing toward the NMES-stimulated (dominant) arm, so they were able to see the stimulated muscle in the mirror but not the contralateral (non-dominant) arm. The subjects were asked to concentrate continuously on the stimulated muscle in the mirror during the stimulation training (Figure 1). The NMES group had the same stimulation training but without the MVF. After each stimulation set, the research staff asked the participants if they could tolerate a higher stimulation amplitude for the subsequent set. If they agreed and could withstand it, the stimulation amplitude was raised by 1 mA for the next set. After completing all 20 sets of NMES, the subjects were asked to rate the overall discomfort and level of physical exertion during the training session by marking the VAS and 6-20 RPE scale, respectively. The mean NMES current intensity for the initial session (Session 1) for both NMES and NMES + MVF groups was 16.5 ± 10.7 mA. The CON group did not receive any training but were asked to maintain their normal daily activities for 3 weeks. At least two days after the last training session, post-tests were conducted in the exact same manner and order as they were conducted during the pre-testing.

### 2.4. Measurements

#### 2.4.1. Elbow Flexion Isometric Strength with Twitch Interpolation

The procedure for the elbow flexion isometric strength testing was similar to the familiarization process. After a brief warm-up, the participants were instructed to perform three 5-s maximal isometric elbow flexions with one minute of rest between each contraction, exerting as much force as possible. The researchers counted down from 3 and verbally encouraged the subjects with “pull, pull, pull” until it was time for them to relax. The twitch interpolation technique was applied during the 2nd and 3rd contractions by delivering a paired-pulse stimulus (100 Hz, 200 μs pulse-width) around 2 s into the maximal contraction, followed by another paired pulse stimulus around 3 s after the contraction, when the participants were fully relaxed [21]. The stimulation current intensity was set at 120% of the current intensity recorded from the twitch interpolation amplitude determination procedure during familiarization, with a range of 84–180 mA for the non-dominant arm and 108–156 mA for the dominant arm. The force signals during all the maximal contractions were collected and sampled at 2222 Hz with a wireless system (NeuroMap System, Delsys Inc., Natick, MA, USA) and then stored in a laboratory computer for further analyses. The maximum force output was calculated for the peak 1-s window of each maximal contraction, and the average of the three maximal force outputs were recorded as the maximal isometric strength. For the maximal contractions with the twitch interpolation, the superimposed twitch force and resting twitch force (Figure 2) were first determined and calculated for each maximal contraction. The average twitch values between the second and third maximal contractions were then used to compute the voluntary activation (VA) percentage using the following formula: VA (%) = (1 − superimposed twitch force/resting twitch force) × 100% [24].

#### 2.4.2. RPE

For this study, a 6–20 Borg RPE scale was utilized [25]. The researchers instructed the participants as follows: “Think of sitting calmly as a 6-No exertion at all and performing a maximal muscle contraction such as our elbow flexion maximal isometric contraction as a 20. How hard did your muscles work during the electrical stimulation?” After completing the 20 sets of NMES for each training session, the subjects were asked to indicate a number from the RPE scale.

#### 2.4.3. NMES-Induced Discomfort—VAS

For the discomfort level assessment in this study, a 100-mm Visual Analog Scale (VAS) was employed. The scale ranged from “no discomfort at all” on the left side to “unbearable discomfort or pain” on the right side. After completing the 20 sets of NMES for each training session, the participants were instructed to mark a vertical line on the VAS scale to indicate their discomfort level [26].

#### 2.4.4. NMES-Evoked Force

To calculate the NMES-evoked force, the force signal data from the entire 20-s stimulation period for each of the 20 training sets in each training session were averaged to obtain the absolute evoked force. Thus, each training session yielded 20 sets of averaged NMES-evoked force. To examine the trend of the NMES-evoked force throughout the entire 9 training sessions, these values (20 sets of force for each training session) were further averaged, so each training session has a mean NMES-evoked force. This number was then divided by the baseline maximal isometric strength to calculate the relative evoked force level (% maximal isometric strength) [23].

### 2.5. Statistical Analyses

All results were presented as mean ± standard deviation (SD). The Shapiro-Wilks test was used to check and confirm that the dependent variables were normally distributed. The baseline dependent variables (isometric strength, voluntary activation level, resting twitch) for both dominant and non-dominant arms were examined via the one-way analysis of variance (ANOVA) tests for the three groups [NMES + MVF, NMES, CON], where no significant differences were found. Additionally, potential gender differences at the baseline were also checked via independent samples *t*-tests, where only the baseline isometric strength was significantly different between men and women. Thus, we used one-way ANOVA to examine the percent change (%Δ) of the isometric strength between baseline and post-intervention among three groups for both non-dominant and dominant arms. Separate two-way (time [Baseline, Post] × group [NMES + MVF, NMES, CON]) mixed factorial ANOVA tests were used to examine the potential changes of other dependent variables for both non-dominant and dominant arms. Additionally, mean NMES current intensity, RPE, NMES-induced discomfort, and relative evoked force during all 9 training sessions were also examined (only for the experimental groups) using two-way mixed factorial ANOVA tests (time [Training session 1, Training session 2, … Training session 9] × group [NMES + MVF, NMES]). The partial *η*^2^ statistic was provided for all the repeated measure comparisons, with values of 0.01, 0.06, and 0.14 representing small, medium, and large effect sizes, respectively [27]. In addition, Cohen’s *d* [27] was calculated for paired comparisons, with 0.2, 0.5, and 0.8 corresponding to small, medium, and large effect sizes, respectively. All the statistical tests were conducted using statistical software (IBM SPSS Statistics 26.0; IBM, Armonk, NY, USA) with an alpha set at 0.05.

## 3. Results

### 3.1. Contralateral Limb Muscle (Untrained)

Table 1 shows the mean values of all the dependent variables (isometric strength, voluntary activation level, resting twitch) of the contralateral elbow flexor muscle for all three groups before and after the interventions or control. The one-way ANOVA for the %Δ of the isometric strength showed no significant difference among groups (*F* = 0.672, *p* = 0.529). For the voluntary activation and resting twitch, the two-way repeated measures ANOVA tests showed no significant interactions, no main effects for time, and no main effects for the group for both dependent variables.

### 3.2. Unilateral Limb Muscle (Trained)

Table 2 shows the mean values of all the dependent variables of the unilateral elbow flexor muscle for all three groups before and after the interventions or control. The one-way ANOVA for the %Δ of the isometric strength indicated a significant difference among the three groups (*F* = 4.014, *p* = 0.046). The Post Hoc LSD tests showed a significant difference between the NMES + MVF and the CON (NMES + MVF vs. CON = 6.31 ± 4.56% vs. −4.04 ± 3.85%, *p* = 0.022), as well as a significant difference between the NMES and the CON (NMES vs. CON = 4.72 ± 8.97% vs. −4.04 ± 3.85%, *p* = 0.046). For the voluntary activation and resting twitch, the two-way repeated measures ANOVA tests showed no significant interactions, no main effects for time, and no main effects for the group for both dependent variables.

Throughout the 9 training sessions, the two-way ANOVA found no significant interaction (*F* = 0.716, *p* = 0.537, partial *η*^2^ = 0.093) or main effects for time (*F* = 1.685, *p* = 0.209, partial *η*^2^ = 0.194) and group (*F* = 0.309, *p* = 0.596, partial *η*^2^ = 0.042) for the NMES current intensity. For both RPE and NMES-induced discomfort (VAS), no significant interactions were found, but there were main effects for the group (RPE: *F* = 6.631, *p* = 0.037, partial *η*^2^ = 0.486; VAS: *F* = 5.633, *p* = 0.049, partial *η*^2^ = 0.446), showing the NMES + MVF group had overall higher RPE and VAS than the NMES group during the training sessions (Figure 3b,c). For the NMES-evoked relative force, there was a significant main effect for time (*F* = 7.118, *p* < 0.001, partial *η*^2^ = 0.504), showing an increased relative evoked force level over time (Figure 3d).

## 4. Discussion

The main purpose of this pilot study was to examine the potential training adaptations of a unilateral neuromuscular electrical stimulation with (NMES + MVF) and without (NMES) mirror visual feedback on the trained (unilateral) and untrained (contralateral) elbow flexor muscles. To our knowledge, this is the first study to examine the combining training effects of NMES and MVF. The main findings of the study are: (1) The cross-education effects were not observed in the contralateral untrained limb muscles for both experimental groups; (2) The unilateral trained muscle became stronger in both experimental groups when compared to the control group, but adding mirror visual feedback provides no superior effect than the NMES only; (3) During training, combining the mirror visual feedback amplified the NMES-induced discomfort and perceived physical exertion when compared to the NMES only group; (4) Throughout training sessions, both experimental groups’ NMES-evoked force gradually increased.

The potential cross-education effects of NMES training have been examined in several studies with healthy subjects [18,28,29,30,31,32,33,34,35]. According to a recent review [17], the majority of the studies investigating the chronic effects of unilateral electrical stimulation have shown some cross-education effects. Barss et al. [28] did not observe the cross-education effect in their study, but it is important to mention that the cutaneous electrical stimulation was applied on the superficial radial nerve rather than the muscles (in the current and many other investigations). Comparing our experiment to the literature, the training intensity (electrical stimulation-evoked force level) seems to be an important factor influencing the results. The current training intensity (across both experimental groups) progressively increased, ranging from 4.2% to 13.5% of the isometric strength. Lai et al. [34] had subjects go through a 3-week high- (50% maximal isometric strength) and low-intensity (25% maximal isometric strength) unilateral knee flexor muscle stimulation and reported that the high- and low-intensity groups had the contralateral muscle strength increased by 24% and 18%, respectively. In a different study, applying stimulation at an intensity of 65% isometric strength also yielded a 21% strength increase in the contralateral limb muscle [33]. Thus, as suggested in a recent review [36], unilateral exercise intensity is one of the key factors in inducing cross-education. Additionally, our three-week training sessions were also relatively short when compared to previous investigations (e.g., 6 weeks) [17]. Such a short training period may only be enough to induce neural but not muscular adaptations in the trained muscles.

In a recent study [19], 10 sessions of NMES at the intensity of 20% isometric strength did not induce strength improvement in the contralateral limb muscle. Instead, motor imagery training was effective in inducing cross-education. Thus, the authors suggested that activating the cortical motor regions is more important than only activating the muscle to induce cross-education. To test the hypothesis that illusionary MVF promotes alterations in the contralateral motor cortical regions to augment the cross-education effect, the MVF was superimposed on the unilateral NMES training in the current study. However, cross-education effects were not observed in the current study. In addition to the lower training intensity in the current study than that in Bouguetoch et al. [19], the difference between using MVF and motor imagery interventions is worth the discussion. The subjects in the motor imagery group from Bouguetoch et al. [19] did 40 motor imagery training of maximal isometric plantar flexions of the right leg with a 6-s on 6-s off tempo. In the current study, the subjects in the NMES + MVF group did not receive specific instructions other than to “focus on the stimulated biceps muscle from the mirror”. Thus, it is possible that the illusionary visual feedback in the current investigation might not be potent enough to activate the contralateral motor cortical region.

When examining the unilateral trained muscle, the current pilot data suggested that subjects in both NMES + MVF and NMES groups experienced unilateral elbow flexion strength improvements following a 9-session training. However, our data does not support that adding the MVF to the NMES can induce any significant superior effects (*p* > 0.05, *d* = 0.22) compared to the NMES only. One interesting finding of this experiment is that the NMES-induced discomfort and perceived physical exertion were significantly greater in the NMES + MVF group than those in the NMES-only group throughout the training sessions. This may be explained by sensory discrepancy due to the illusionary MVF. In the experiment by McCabe et al. [37], nearly 2/3 of the healthy subjects reported sensory changes (e.g., discomfort to mild pain, temperature change, weight change, perceived loss of or additional limbs, etc.) when they performed congruent or incongruent bilateral limb movement while viewing the reflected limb in the mirror. When using MVF, the visual feedback provided by the mirror may be different from the individual’s proprioceptive and tactile feedback, and this sensory mismatch may create a sense of discomfort. Thus, it is possible that adding MVF to NMES in the current study amplified discomfort, thereby altering the physical exertion perception of the NMES-induced involuntary muscle contraction. This difference may account for the small effect of isometric strength improvements between the NMES + MVF and NMES groups. However, the NMES current intensities and NMES-induced force output were not significantly different between the two experimental groups during the training sessions. Both experimental groups experienced gradually increased NMES-evoked force throughout the training, similar to the findings from Neyroud et al. [38]. Thus, adding the MVF did not increase training intensity, which explains the insignificant training effects on unilateral isometric strength for the two experimental groups. Compared to literature that has used NMES as a training intervention [17], our result seems relatively low. As mentioned above, this is likely due to the relatively low NMES-evoked force in the current study. Even though the NMES current intensity was progressively increased due to the improved NMES tolerance throughout the training, the training intensity (NMES-evoked force) in the current study was still lower than those in some previous studies (e.g., 20% of isometric strength). One question that remains unanswered is what exactly the mechanism(s) might be to induce the unilateral trained (stimulated) muscle strength increase. The voluntary activation level (measuring central component) and the resting twitch force (mearing peripheral level) did not change before or after the NMES training (with or without MVF). This suggests that the mechanistic factors influencing the strength were likely at the supraspinal level (e.g., changes in the corticospinal activity).

This pilot study has some limitations. First, without examining the cortical activities using instruments such as EEG and TMS, we can only speculate on the mechanisms involved in the supraspinal site. Second, while training at a maximally tolerable stimulation intensity has some practicality, this does not allow the researchers to deliver the exact same intervention (e.g., reaching a certain level of NMES-evoked force) to all the subjects. Thus, results can be different if setting the NMES-evoked force constant across individuals at a relatively higher level were involved and/or if the intervention period were longer (e.g., 6 weeks). Third, the current investigation suffers from the small sample size. Future research may yield more useful information if it includes enough healthy and/or clinical subjects.

## 5. Conclusions

In conclusion, three weeks of unilateral upper limb muscle NMES training with or without illusionary MVF did not induce any cross-education effects on the contralateral limb muscle. It is possible that increasing both unilateral training intensity (NMES stimulation amplitude) and training volume (number of training sessions) may alter the results. At the maximally tolerable stimulation current intensity, the unilateral muscle training effect on isometric strength was evident. However, adding the MVF to NMES training only seemed to have a small magnitude effect on strength increase. This training adaptation was likely due to changes above the spinal level. Additionally, adding MVF to NMES seemed to augment the overall NMES-induced discomfort and the perception of physical exertion. This can make MVF disadvantageous when implementing it in an exercise intervention. Future investigations should focus on examining the mechanistic factors leading to the NMES training-induced neuromuscular changes, as well as examining possible interventions on clinically-relevant subjects (e.g., stroke).

## Figures and Tables

**Figure 1 ijerph-20-03755-f001:**
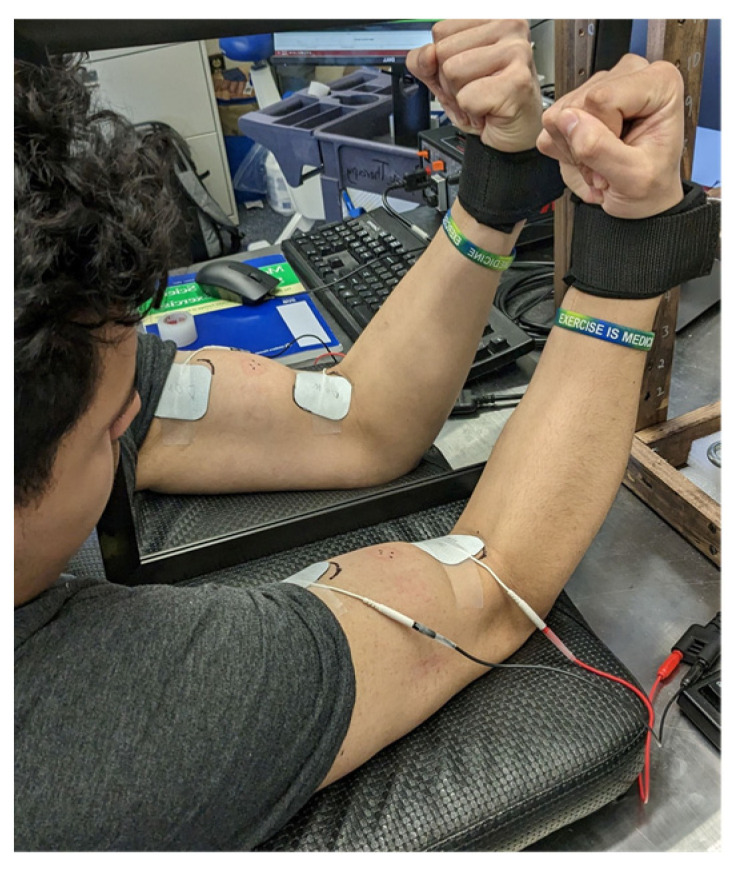
Experimental setup for a subject from the NMES + MVF group.

**Figure 2 ijerph-20-03755-f002:**
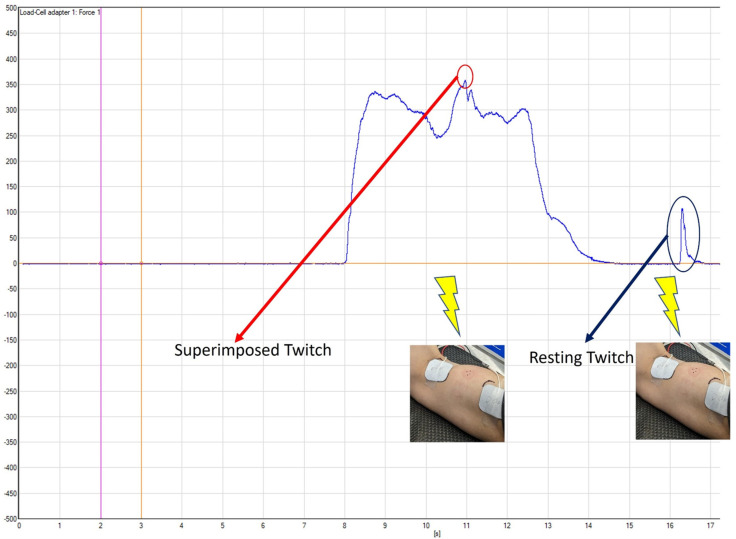
The time-varying force curve from a representative subject’s maximal isometric contraction with the twitch interpolation.

**Figure 3 ijerph-20-03755-f003:**
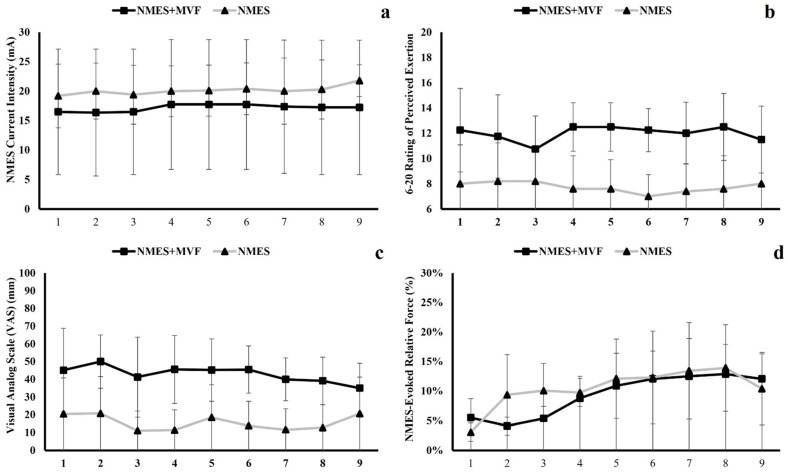
Changes of the mean stimulation current intensity (**a**), RPE (**b**), NMES-induced discomfort (VAS) (**c**), and NMES-evoked relative force (**d**) for both NMES + MVF and NMES groups across the 9 training sessions.

**Table 1 ijerph-20-03755-t001:** Mean and standard deviation (SD) for the isometric strength, voluntary activation level, and resting twitch of the non-dominant (untrained) elbow flexor muscle for all groups.

	Isometric Strength (N)		Voluntary Activation (%)		Resting Twitch (N)	
	*Baseline*	*Post*	*Baseline*	*Post*	*Baseline*	*Post*
NMES + MVF	273.3 ± 94.1	278.4 ± 89.2	90.11 ± 5.17	94.62 ± 3.18	52.6 ± 34.2	56.1 ± 31.1
NMES	316.1 ± 54.4	321.8 ± 54.5	87.66 ± 9.05	94.58 ± 3.88	45.5 ± 28.2	39.4 ± 26.0
CON	300.5 ± 119.2	299.9 ± 90.6	95.88 ± 3.24	97.16 ± 1.80	63.7 ± 40.2	61.2 ± 38.7

NMES + MVF: Neuromuscular electrical stimulation combined with mirror visual feedback group; NMES: Neuromuscular electrical stimulation group; CON: Control group.

**Table 2 ijerph-20-03755-t002:** Mean and standard deviation (SD) for the isometric strength, voluntary activation level, and resting twitch of the dominant (trained) elbow flexor muscle for all groups.

	Isometric Strength (N)			Voluntary Activation (%)		Resting Twitch (N)	
	*Baseline*	*Post*	*% Change*	*Baseline*	*Post*	*Baseline*	*Post*
NMES + MVF	270.6 ± 101.4	285.1 ± 99.2	6.31 ± 4.56% *****	91.61 ± 4.54	90.20 ± 8.67	54.1 ± 20.5	54.8 ± 24.1
NMES	304.7 ± 56.2	316.5 ± 46.9	4.72 ± 8.97% *****	93.34 ± 2.54	92.97 ± 3.60	56.9 ± 21.1	56.5 ± 35.9
CON	310.3 ± 129.9	297.7 ± 124.2	−4.04 ± 3.85%	95.63 ± 1.95	94.69 ± 2.84	57.9 ± 22.2	55.0 ± 22.2

NMES + MVF: Neuromuscular electrical stimulation combined with mirror visual feedback group; NMES: Neuromuscular electrical stimulation group; CON: Control group; ***** Significant difference between the experimental group (NMES + MVF or NMES) and the control group.

## Data Availability

The data presented in this study are available on request from the corresponding author. The data are not publicly available due to the ongoing investigation not being completed.

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
