# Peer review of "Effects of Unilateral Neuromuscular Electrical Stimulation with Illusionary Mirror Visual Feedback on the Contralateral Muscle: A Pilot Study"

_ijerph, 2023, doi:10.3390/ijerph20043755_

Round 1

Reviewer 1 Report

In the submitted manuscript by Xin Ye et al., the authors examine the effects of cross-education of unilateral NMES (neuromuscular elect. stim.) on the dominant arm. They had 3 groups of participants: NMES only, NMES + Mirror Feedback, and control. They found that adding mirror feedback increases discomfort and the sense of exertion. NMES with and without mirror feedback does not induce cross-education in the contralateral, untrained limb. The article was written in a straightforward manner, easy to read.

Questions/feedback:

1)       I strongly suggest placing two figures for the purpose of the journal audience: First figure is showing the experimental setup with a participant; the second figure shows typical time-varying force measurement from the maximal isometric twitch interpolation technique of a representative participant.

2)       Line 171, why did the authors increase the stimulation amplitude again (given that this amplitude is already the maximum tolerable one)?

3)       Line 176, after at least 2 days post-training, does the gap potentially cause the null effect of cross-education?

4)       Do the authors think the lack of effect can be due to the insufficient number of training sessions? In suggesting a future direction in the Conclusion, why didn't the authors consider adding the sample size (given the potential treatment application in neurological injury/stroke)?

5)       Please add 1-2 sentences discussing the neural mechanisms why cross-education is possible, e.g. the carpus-callosum connecting both cortical hemispheres.

Author Response

Responses to comment from Reviewer #1

Following are the comments from Reviewer 1, as well as our point-by-point responses (red, italic) to the comments. In addition, we have revised our manuscript main text based on the comments from the reviewers. Specifically, we highlighted the revised parts in the revised manuscript.

Comments and Suggestions for Authors

In the submitted manuscript by Xin Ye et al., the authors examine the effects of cross-education of unilateral NMES (neuromuscular elect. stim.) on the dominant arm. They had 3 groups of participants: NMES only, NMES + Mirror Feedback, and control. They found that adding mirror feedback increases discomfort and the sense of exertion. NMES with and without mirror feedback does not induce cross-education in the contralateral, untrained limb. The article was written in a straightforward manner, easy to read.

Thank you very much for the positive comments. We have addressed your following comments, as well as revised our manuscript accordingly. We hope the revised manuscript is now significantly improved.

Questions/feedback:

1)       I strongly suggest placing two figures for the purpose of the journal audience: First figure is showing the experimental setup with a participant; the second figure shows typical time-varying force measurement from the maximal isometric twitch interpolation technique of a representative participant.

Thank you for the suggestion. We now have added two figures (Figure 1 and Figure 2) to illustrate the experimental setup and the representative ITT force signals.

2)       Line 171, why did the authors increase the stimulation amplitude again (given that this amplitude is already the maximum tolerable one)?

Thank you for the question. Based on our observation and previous NMES interventions, many individuals’ tolerance to the NMES can increase over time. We, therefore, constantly encouraged our participants to try higher amplitude to achieve their true maximal tolerance. Theoretically, the higher the amplitude, the greater NMES-elicited force, which may make the training effect more obvious.

3)       Line 176, after at least 2 days post-training, does the gap potentially cause the null effect of cross-education?

Thank you for this question. The main reason we had the subjects come back at least two days after the last training session is that we did not want the last training session to influence the post-intervention testing. Even though the training intensity (%MVC) was light, some subjects reported that the NMES training was intensive, and they felt their muscle were sore or “worked”. Therefore, it is important to provide a small gap to make sure their muscles were not influenced by training-induced fatigue. Indeed, if we tested the cross-education effect (contralateral arm) immediately after the last training session, it is possible we might have observed something. But this effect was more likely induced by the acute NMES rather than chronic training.

4)       Do the authors think the lack of effect can be due to the insufficient number of training sessions? In suggesting a future direction in the Conclusion, why didn't the authors consider adding the sample size (given the potential treatment application in neurological injury/stroke)?

Thank you for the suggestion. We agree with you that in addition to the training intensity, training volume (number of training sessions) is also an important factor. And yes, we are taking your advice to suggest adding sample size, and examining the clinically relevant patients. See lines 395-397, 411-412.

5)       Please add 1-2 sentences discussing the neural mechanisms why cross-education is possible, e.g. the carpus-callosum connecting both cortical hemispheres.

Thank you for the suggestion. We have added the contents of possible mechanisms to the beginning of the Introduction: lines 36-40

Reviewer 2 Report

REVIEWER Comments:

Congratulations on choosing a good topic. However, you need to check and correct some parts of the thesis for completeness.

 Methods

1.      The methods are very specific and detailed.

2.      It is suggested that the authors add information on the initial stimulus size for MNES to the Methods section (lines 159–178).

 Results

1.      It is suggested that the authors provide the statistical value and the significant difference together in Tables 1 and 2.

Discussion and Conclusions:

1.      The authors ought to briefly discuss why adding MVF to NMES increases NMES discomfort.

Author Response

Responses to comment from Reviewer #2

Following are the comments from Reviewer 1, as well as our point-by-point responses (red, italic) to the comments. In addition, we have revised our manuscript main text based on the comments from the reviewers. Specifically, we highlighted the revised parts in the revised manuscript.

Comments and Suggestions for Authors

Congratulations on choosing a good topic. However, you need to check and correct some parts of the thesis for completeness.

Thank you very much for the positive comments. We have addressed your following comments and revised our manuscript accordingly. We hope the revised manuscript is now significantly improved.

Methods

  1. The methods are very specific and detailed.
  2. It is suggested that the authors add information on the initial stimulus size for MNES to the Methods section (lines 159–178).

Thank you! We now have added the initial stimulus size to this paragraph (Lines 181-182): “The mean NMES current intensity for the initial session (Session 1) was 16.5 ± 10.7 mA”

 Results

  1. It is suggested that the authors provide the statistical value and the significant difference together in Tables 1 and 2.

Thank you for the suggestion. For Table 1 (contralateral limb variables, there were no significant findings for all variables), so we did not add any statistical values with significant differences. We did report the F-values and p-values in the main text (Lines 266-267). We have now added the statistical values to Table 2, specifically for the isometric strength.

Discussion and Conclusions:

  1. The authors ought to briefly discuss why adding MVF to NMES increases NMES discomfort.

Thank you for the suggestion. We believe the amplified discomfort may come from the mirror visual feedback-induced sensory discrepancy. We now have added the discussion about the NMES+MVF-induced discomfort (Lines 361-369).

Reviewer 3 Report

The manuscript represents a research with the aim to examine the cross-education effects of unilateral muscle neuromuscular electrical stimulation training combined with illusionary mirror visual feedback. These are my comments and suggestions:

Abstract:

Please check the manuscript for typograhical errors (e.g. measurem+ents). If the changes were significant, include p values.

Introduction:

Introduction is nicely written, with adequate theoretical background and clearly stated aim of the study.

Methods:

If the study had randomized controlled design it should have been prospectively registered. Did you register the study protocol? Despite this is a pilot study, the sample is small and should be justified. How did you estimate your sample size? Please, include references for the validity of muscle twitch interpolation amplitude determination. Please, justify relatively short training period (3 weeks), i.e. its methodological adequacy to induce muscular changes. Photograph of the training regime using mirror would be useful. Please, provide reference for NMES-induced discomfort VAS. 

Results:

No comments, well presented.

Discussion:

I suggest to avoid repeating results (numeric data) in the Discussion. Please, explain why your study is important and novel. Add few sentences regarding its clinical significance. Scientific contribution of the study should be more emphasized.

Author Response

Responses to comment from Reviewer #3

Following are the comments from Reviewer 1, as well as our point-by-point responses (red, italic) to the comments. In addition, we have revised our manuscript main text based on the comments from the reviewers. Specifically, we highlighted the revised parts in the revised manuscript.

Comments and Suggestions for Authors

The manuscript represents a research with the aim to examine the cross-education effects of unilateral muscle neuromuscular electrical stimulation training combined with illusionary mirror visual feedback. These are my comments and suggestions:

Thank you very much for the comments. We have addressed your following comments, as well as revised our manuscript accordingly. We hope the revised manuscript is now significantly improved.

Abstract:

Please check the manuscript for typograhical errors (e.g. measurem+ents). If the changes were significant, include p values.

Thank you for pointing these out. The typos are now checked and corrected. We’ve also added the p-value (< 0.05) to the abstract. Because there were two p-values (between NMES+MVF vs. Control, and between NMES vs. control), to limit our words under the required number (200), we used p < 0.05 instead of the exact number (can be found in the results section).

Introduction:

Introduction is nicely written, with adequate theoretical background and clearly stated aim of the study.

Methods:

If the study had randomized controlled design it should have been prospectively registered. Did you register the study protocol? Despite this is a pilot study, the sample is small and should be justified. How did you estimate your sample size? Please, include references for the validity of muscle twitch interpolation amplitude determination. Please, justify relatively short training period (3 weeks), i.e. its methodological adequacy to induce muscular changes. Photograph of the training regime using mirror would be useful. Please, provide reference for NMES-induced discomfort VAS. 

Thank you for the comments on the Methods section. The study protocol was not registered.

Regarding the sample size estimation, there was no previous study examined the combination effects of NMES and MVF, we therefore only used the NMES only to estimate the sample size, based on the work of Bouguetoch et al. (2021). Does partial activation of the neuromuscular system induce cross-education training effect? Case of a pilot study on motor imagery and neuromuscular electrical stimulation. With the large effect size for the change in NMES-induced trained limb strength gain (f = 0.4), power = 0.8, the current investigation requires 17 subjects per group.

We have added references for the validity of muscle twitch interpolation amplitude determination (Reference 20).

Regarding the relatively short training period, one of our reasons for choosing a short 3-week is because some recent NMES training studies also used this time period (Bouguetoch et al. 2021: 2 weeks 10 training sessions; Neyroud et al. 2019: 3 weeks 9 training sessions). Additionally, training adaptations beyond 3 or 4 weeks usually start to involve muscular changes, our main purpose was to examine the possible neural adaptations, thus we chose to limit the training period to 3 weeks. Indeed, this may be the main reason that we did not observe any cross-education effects. We have added this as a possible limitation to our discussion (lines 334-337, and 395).

We have added a figure (Figure 1) to illustrate the experimental setup.

Lastly, we have added a reference for the NMES-induced discomfort (Reference 24).

Results:

No comments, well presented.

Discussion:

I suggest to avoid repeating results (numeric data) in the Discussion. Please, explain why your study is important and novel. Add few sentences regarding its clinical significance. Scientific contribution of the study should be more emphasized.

Thank you for the suggestions. We’ve revised the Discussion, and we hope the current discussion improved the manuscript: Lines 309-310 (significance of the study); Lines 372 and 378 (numeric data removed); Line 409 and 411-412 (clinical significance).